# The *Ralstonia solanacearum* Type III Effector RipAW Targets the Immune Receptor Complex to Suppress PAMP-Triggered Immunity

**DOI:** 10.3390/ijms25010183

**Published:** 2023-12-22

**Authors:** Zhi-Mao Sun, Qi Zhang, Yu-Xin Feng, Shuang-Xi Zhang, Bi-Xin Bai, Xue Ouyang, Zhi-Liang Xiao, He Meng, Xiao-Ting Wang, Jun-Min He, Yu-Yan An, Mei-Xiang Zhang

**Affiliations:** 1College of Life Sciences, Shaanxi Normal University, Xi’an 710119, China; szm@snnu.edu.cn (Z.-M.S.); zhangqi00@snnu.edu.cn (Q.Z.); yxfeng@snnu.edu.cn (Y.-X.F.); sxzhang2022@snnu.edu.cn (S.-X.Z.); bx_bai@snnu.edu.cn (B.-X.B.); 18652075802@163.com (X.O.); 15996245235@163.com (X.-T.W.); hejm@snnu.edu.cn (J.-M.H.); 2Chinese Academy of Agricultural Sciences, Qingdao 266101, China; xiaozhiliang@caas.cn (Z.-L.X.); menghe@caas.cn (H.M.)

**Keywords:** bacterial wilt, *Ralstonia solanacearum*, RipAW, PAMP-triggered immunity, immune receptor

## Abstract

Bacterial wilt, caused by *Ralstonia solanacearum*, one of the most destructive phytopathogens, leads to significant annual crop yield losses. Type III effectors (T3Es) mainly contribute to the virulence of *R. solanacearum*, usually by targeting immune-related proteins. Here, we clarified the effect of a novel E3 ubiquitin ligase (NEL) T3E, RipAW, from *R. solanacearum* on pathogen-associated molecular pattern (PAMP)-triggered immunity (PTI) and further explored its action mechanism. In the susceptible host *Arabidopsis thaliana*, we monitored the expression of PTI marker genes, flg22-induced ROS burst, and callose deposition in *RipAW*- and *RipAW^C177A^*-transgenic plants. Our results demonstrated that RipAW suppressed host PTI in an NEL-dependent manner. By Split-Luciferase Complementation, Bimolecular Fluorescent Complimentary, and Co-Immunoprecipitation assays, we further showed that RipAW associated with three crucial components of the immune receptor complex, namely FLS2, XLG2, and BIK1. Furthermore, RipAW elevated the ubiquitination levels of FLS2, XLG2, and BIK1, accelerating their degradation via the 26S proteasome pathway. Additionally, co-expression of FLS2, XLG2, or BIK1 with RipAW partially but significantly restored the RipAW-suppressed ROS burst, confirming the involvement of the immune receptor complex in RipAW-regulated PTI. Overall, our results indicate that RipAW impairs host PTI by disrupting the immune receptor complex. Our findings provide new insights into the virulence mechanism of *R. solanacearum*.

## 1. Introduction

Plants have evolved a sophisticated immune system during their prolonged struggle with plant pathogens. Simultaneously, successful plant pathogens have developed corresponding strategies to suppress plant immunity. The plant immune system comprises two main branches: pathogen-associated molecular pattern (PAMP)-triggered immunity (PTI) and effector-triggered immunity (ETI) [1]. PTI is activated by plasma-membrane-localized pattern recognition receptors (PRRs), which recognize pathogen PAMPs early in the invasion process and initiate downstream immune responses, such as a reactive oxygen species (ROS) burst, callose deposition, and calcium (Ca^2+^) influx [2].

As the first layer of plant immunity, PTI plays a crucial role in monitoring and preventing the invasion of various plant pathogens [3]. PTI consists of several modules, with the PRR complex serving as a pivotal component that acts as a sensor [4]. The intact immune receptor complex is vital for PTI activation, and various immune complexes have been identified. For instance, the co-receptor BRI1-associated receptor kinase (BAK1) interacted with flagellin-sensitive 2 (FLS2) or brassinosteroid-insensitive 1 (BRI1) to form the receptor complex, recognizing the corresponding flagellin or brassinosteroid. FLS2 can also form a complex with receptor-like cytoplasmic kinases (RLCKs) and G-proteins, such as *botrytis*-induced kinase 1 (BIK1) [5,6,7]. Upon sensing stimuli, members of the immune receptor complex undergo modifications, causing conformational changes, disassociation from the complex, and downstream signal transmission. For instance, upon PAMP sensing, BIK1 is monoubiquitinated following phosphorylation, released from the FLS-BAK1 complex [6], and activates respiratory burst oxidase homolog D (RBOHD) [8]. The Gα proteins extra-large guanine nucleotide-binding protein 2 (XLG2) and guanine nucleotide-binding protein alpha-1 (GPA1) activate mesophyll immunity and stomatal immunity upon immune signaling stimuli [7,9,10]. Meanwhile, BIK1 is well-known as an immunity hub, conveying signals from multiple receptors [6,11]. In addition to being regulated by receptors, BIK1 is also modulated by G-proteins and calcium-dependent protein kinase. Heterotrimeric G proteins stabilize BIK1 by inhibiting the U-box domain-containing protein (PUB25/26) activity, while calcium-dependent protein kinase (CPK28) shows an opposite function [12]. The G protein-CPK28 regulatory module maintains the homeostasis of BIK1 and plant immunity [12]. Therefore, these components of the immune receptor complex are crucial for signal sensing and transduction.

To counteract the plant immune system, plant pathogens secrete a variety of effectors to interfere with plant immunity. Immune activation initiates a cascade of downstream immune responses. Blocking the recognition of PAMPs at the receptor level is a highly effective strategy for pathogens to suppress plant immunity. Numerous cases have been reported that bacterial effectors target plant immune receptor complexes to suppress PTI. For example, AvrPto and AvrPtoB from *Pseudomonas syringae* target the co-receptor BAK1, disrupting receptor-ligand recognition and blocking signal transduction [13]. AvrPto also targets the receptor of flagellin and EF-TU, FLS2, and EFR, disrupting the perception of pathogens to suppress PTI [14]. AvrPtoB, which has an E3 ligase activity, ubiquitinates Fen, promoting its degradation and enhancing *P. syringae* virulence [15]. Moreover, bacterial effectors can also target the downstream signal transduction modules of the immune receptor complex. Xoo1488 from *Xanthomonas oryzae* targets OsRLCK185 to suppress chitin-induced PTI mediated by chitin elicitor receptor kinase 1 (CERK1) [16]. Xoo1488 interacts with OsRLCK185 to inhibit the immune signal transduction from CERK1. Additionally, the protease effector AvrPphB can cleave the PBS1-like (PBL) kinase, including PBL1, PBL2, and BIK1, to suppress PTI signal transduction [17]. Therefore, due to the essential role of the immune receptor complex in plant immune activation, it is a common and effective strategy for phytopathogen effectors to impair plant immunity by disrupting the immune receptor complex.

Bacterial wilt disease, caused by *Ralstonia solanacearum*, leads to significant annual crop yield losses, and *R. solanacearum* is recognized as one of the most devastating bacterial phytopathogens [18]. The type III effectors (T3Es) of *R. solanacearum* play a major role in its virulence [19,20,21]. For example, the well-characterized RipAC has been reported to target the immunity hub, a suppressor of the G2 allele of skp1 (SGT1), to enhance *R. solanacearum* virulence [19,22]. In this study, we focus on RipAW of *R. solanacearum*, which possesses a novel E3 ubiquitin ligase (NEL) domain. It has been demonstrated that RipAW exhibits E3 ubiquitin ligase activity, with this activity depending on the cysteine 177 residue [23]. Niu et al. found that RipAW induced a strong hypersensitive response (HR) and programmed cell death (PCD) in *Nicotiana benthamiana* and *Nicotiana tabacum*, relying on its E3 ligase activity [24]. Ouyang et al. further demonstrated that the E3 ligase activity of RipAW is not essential for the induction of plant defense [25]. Interestingly, RipAW has also been shown to impair PTI, although it can activate ETI in *N. benthamiana* [23]. However, how RipAW interferes with plant immunity and what its host target is remain unknown.

In this study, to confirm the effect of RipAW on PTI, we conducted a comprehensive study on *RipAW*-transgenic *Arabidopsis thaliana*. By monitoring the expression of the PTI marker gene, detecting flg22-induced ROS burst, and assessing callose deposition, we confirmed that RipAW reduces plant resistance to pathogens by interfering with PTI. Significantly, our investigation revealed that RipAW suppresses PTI in plants by associating with the immune receptor complex. RipAW elevated the ubiquitination levels of FLS2, XLG2, and BIK1, reducing their protein accumulation. Our findings contribute to a deeper understanding of the pathogenic mechanism of *R. solanacearum*.

## 2. Results

### 2.1. RipAW Impairs Plant Resistance to Different Pathogens in Arabidopsis thaliana

RipAW, a NEL T3E of *R. solanacearum*, has been shown to trigger ETI on *N. benthamiana* [24] while also exhibiting the ability to suppress PTI through its E3 ubiquitin ligase activity [23]. To clarify the effect of RipAW, we further investigated its function of RipAW in the host plant *A. thaliana*. We generated *RipAW*- and *RipAW^C177A^*-transgenic *A. thaliana* and found that the expression of *RipAW* or *RipAW^C177A^* did not obviously affect the plant’s developmental phenotype (Figure 1A). Then, we evaluated the disease resistance of these transgenic plants by inoculating them with two different pathogens, *R. solanacearum* GMI1000 and *Pseudomonas syringae* DC3000. The results revealed a significant increase in the disease index and a decrease in the survival rate in *RipAW*-transgenic plants compared to GFP control plants after soil-drenching inoculation with GMI1000. Conversely, *RipAW^C177A^*-transgenic plants did not show such effects (Figure 1B). This result indicates that *RipAW*-transgenic plants are more susceptible to *R. solanacearum* GMI1000, whereas *RipAW^C177A^*-transgenic plants are not, as compared to GFP control plants. To validate this finding, a Petri-dish inoculation assay was conducted following the method of Cao et al. [26], and the significantly higher population of GMI1000 in *RipAW*-transgenic plants (Figure 1C) supported the result of the soil-drenching inoculation assay. These results highlight that RipAW significantly contributes to the virulence of *R. solanacearum*, and this contribution is dependent on its E3 ligase activity. The notably higher population of DC3000 in *RipAW*-transgenic plants, but not in *RipAW^C177A^*-transgenic plants (Figure 1D), further confirmed the disruption of plant immunity by RipAW and underscored the crucial role of its E3 ligase activity. GMI1000 *HrcV^−^* and DC3000 *HrcC^−^* are strains deficient in the type III secretion system and are widely used to assess host PTI [27,28]. We showed that the bacterial population of these deletion mutants remained higher in *RipAW*-transgenic plants compared to the GFP control and *RipAW^C177A^*-transgenic plants (Figure 1E,F). This result suggests that RipAW likely interferes with plant immunity by suppressing PTI, and its E3 ligase activity is essential for this virulence function.

### 2.2. RipAW Suppresses PAMP-Triggered Immunity in A. thaliana

Given the susceptibility of *RipAW*-transgenic plants to both DC3000 and DC3000 *HrcC^−^*, we speculated that RipAW may disturb plant immunity by suppressing PTI. A series of experiments were conducted to verify whether RipAW suppresses PTI in *A. thaliana*. Firstly, we found that flg22-induced ROS burst was severely suppressed in *RipAW*-transgenic plants compared with the *RipAW^C177A^*- and *GFP*-transgenic plants (Figure 2A). Secondly, the expression of PTI marker genes, *WRKY29* and *FRK1*, was assessed after the flg22 treatment. Before treatment, there was no difference among *RipAW*-, *RipAW^C177A^*-, and *GFP*-transgenic plants. However, after 3 h of flg22 treatment, *RipAW*-transgenic plants showed significantly lower expression of *WRKY29* and *FRK1* (Figure 2B). Moreover. Additionally, we measured callose deposition, another crucial indicator of PTI, in transgenic plants following flg22 treatment. Compared with the water control, callose deposition was strongly induced by 1 μM flg22 treatment in all plants. However, compared with the *GFP* control, callose deposition level was significantly impaired by the expression of *RipAW* but not *RipAW^C177A^* (Figure 2C). These results indicate that RipAW suppresses host PTI through its E3 ligase activity. To further confirm the compromise of PTI in *RipAW*-transgenic plants, a defense priming assay was conducted. Compared with the water control, pretreatment with 1 μM flg22 significantly enhanced host resistance to *P. syringae* DC3000, as reported previously (Figure 2D, [17]). The DC3000 population was decreased to 62.95%, 80.73%, and 62.59% in *GFP*-, *RipAW*-, and *RipAW^C177A^*-transgenic plants, respectively (Figure 2D), suggesting that flg22-triggered immunity is impaired by RipAW but not RipAW^C177A^. This result provides further evidence that RipAW suppresses host PTI in an E3 ligase activity-dependent manner.

### 2.3. RipAW Associates with the Immune Receptor Complex

The immune receptor complex plays a crucial role in initiating PTI [29]. Given that RipAW suppressed flg22-triggered PTI in *A. thaliana*, we investigated whether RipAW disrupts host PTI by targeting the immune receptor complex. To test this hypothesis, we examined the association of RipAW with FLS2, XLG2, and BIK1, three essential members of the immune receptor complex. Initially, a split-luciferase complementation assay (LCA) was conducted, revealing that RipAW associated with FLS2, XLG2, and BIK1 in vivo (Figure 3A). To verify these interactions, bimolecular fluorescence complementation (BiFC) and co-immunoprecipitation (co-IP) assays were performed in *N. benthamiana*. The BiFC assay indicated that RipAW associated with FLS2, XLG2, and BIK1, likely in the plasma membranes (Figure 3B). However, in the co-IP assay, we observed the association of RipAW with FLS2 and BIK1, while the interaction between RipAW and XLG2 was not detected, probably due to the weak protein expression of XLG2 (Figure 3C). These results highlight a robust association of RipAW with FLS2 and BIK1 and a comparatively weak association with XLG2.

Since RipAW possesses ubiquitin ligase activity, we supposed that RipAW could accelerate the degradation of its target proteins. Therefore, we investigated the effect of RipAW on the protein accumulation of FLS2, XLG2, and BIK1. As expected, compared with the GFP control, co-expression with RipAW severely suppressed the protein accumulation of FLS2, XLG2, and BIK1, while RipAW^C177A^ failed to achieve this effect (Figure 3D–F). These results indicate that RipAW decreases the accumulation of FLS2, XLG2, and BIK1 through its E3 ligase activity.

### 2.4. RipAW Ubiquitinates FLS2, XLG2, and BIK1 to Decrease Their Protein Accumulation through the 26S Proteasome

The ubiquitin–26S proteasome system plays a critical role in the regulation of protein homeostasis in plants [30]. We speculated that the RipAW could ubiquitinate its target proteins, thereby accelerating protein degradation via the 26S proteasome pathway. To test this hypothesis, we treated plant leaves with 20 μM proteasome inhibitor MG132 for 8 h after co-expression of FLS2, XLG2, or BIK1 with RipAW and detected protein accumulation by western blot. Under the DMSO control treatment, the protein accumulation of FLS2, XLG2, or BIK1 in leaves co-expressed with RipAW was obviously lower compared to the group co-expressed with RipAW^C177A^. However, this difference was abolished by MG132 treatment (Figure 4A–C). These results indicate that the reduced protein accumulation of FLS2, XLG2, and BIK1 in the presence of RipAW was due to protein degradation via the 26S proteasome. Considering that RipAW^C177A^ failed to decrease the protein accumulation of FLS2, XLG2, and BIK1 (Figure 3D–F), it is reasonable to speculate that FLS2, XLG2, and BIK1 are ubiquitinated by RipAW and subsequently degraded by the 26S proteasome.

To primarily address whether RipAW can ubiquitinate FLS2, XLG2, and BIK1, we conducted an in vivo ubiquitination experiment. FLS2, XLG2, or BIK1 was co-expressed with RipAW in N. benthamiana leaves, followed by treatment with MG132 at 36 hpi, and sampled after an additional 8 h. After immunoprecipitation with anti-FLAG agarose beads, the target proteins were detected by an anti-ubiquitin antibody. The input confirmed the protein expression and equal loading. It was evident that FLS2, XLG2, and BIK1 showed significantly higher levels of ubiquitination when co-expressed with RipAW compared to the control and RipAW^C177A^ (Figure 4D–F). These results provide evidence that RipAW promotes the degradation of FLS2, XLG2, and BIK1 by elevating their ubiquitination levels.

### 2.5. FLS2, XLG2, or BIK1 Can Partially Restore RipAW-Suppressed ROS Burst in N. benthamiana

The above results suggest that RipAW’s suppression of PTI is closely linked to its inhibitory effect on FLS2, XLG2, and BIK1. To validate this, we conducted a complementation experiment. We co-expressed FLS2, XLG2, or BIK1 with RipAW in *N. benthamiana* to assess whether these proteins could restore the flg22-induced ROS burst, which is suppressed by RipAW. Clearly, the expression of FLS2, XLG2, or BIK1 could partially but significantly rescue the RipAW-suppressed ROS burst (Figure 5). This result further confirms that the *Ralstonia* T3E RipAW targets the host immune receptor complex to suppress PTI.

## 3. Discussion

As one of the most destructive bacterial phytopathogens in the world [18], *R. solanacearum* causes serious bacterial wilt disease in various plant species, resulting in great losses in crop yield and quality worldwide every year [31]. The T3Es play crucial roles in plant–*R. solanacearum* interactions [32]. Although several virulence and avirulence T3Es have been characterized [33,34,35,36], efficient strategies against bacterial wilt disease remain a challenge due to the limited understanding of the extensive effector repertoires. RipAW, a NEL effector in *R. solanacearum*, exhibits E3 ligase activity dependent on the cysteine 177 residue [17]. Recognition of RipAW and the relationship between RipAW-triggered ETI and its E3 ligase activity has been clarified [24,25]. In this study, we characterized the virulence function of RipAW in the host plant and explored its virulence mechanism, focusing on interference with the immune receptor complex. Our findings contribute to a deeper understanding of the NEL effector RipAW in plant–*R. solanacearum* interactions.

As a T3E, RipAW has been suggested to promote the infection of *R. solanacearum*. Indeed, previous reports indicate that RipAW inhibited the expression of PTI-related genes in *N. benthamiana*, although it can also trigger ETI responses in *N. benthamiana* [17]. In this study, we systematically characterized RipAW in the model plant *A. thaliana*, a susceptible host for *R. solanacearum* GMI1000, to clarify its role in suppressing host PTI. In addition to confirming the previous findings that RipAW decreased the expression of PTI marker genes and flg22-induced ROS burst, we also illustrated that other PTI-related immune responses, such as callose deposition, were also weakened by RipAW. A significantly higher bacterial population of both wild-type DC3000 or GMI1000 and their mutants, DC3000 *HrcC^−^* or GMI1000 *HrcV^−^*, was observed in *RipAW*-transgenic plants but not in *RipAW^C177A^*-transgenic plants. This result indicates that RipAW expression suppresses plant resistance to bacterial pathogens regardless of the presence of the type III secretion system, providing further evidence that RipAW contributes to *R. solanacearum* virulence by interrupting host PTI. RipAW^C177A^, the E3 ligase mutant of RipAW [23], lost this function, confirming that the E3 ligase activity is required for RipAW to interfere with host PTI.

PTI, as the first layer of plant immunity, is triggered in the process of pathogen invasion. Subsequently, immune signals rapidly expand, accompanied by a series of downstream immune responses, including ROS burst, Ca^2+^ influx, and mitogen-activated protein kinase (MAPK) cascades [2,37]. Therefore, blocking plant immunity at the pathogen perception stage is a common and efficient strategy for pathogens to promote infection. Increasing reports have shown that phytopathogens employ effectors, such as AvrPtoB from *P. syringae* [15] and Xoo1488 from *X. oryzae* [16], to interfere with the immune receptor complex and block PTI. However, to date, no information is available on whether this strategy exists in *R. solanacearum*. Here, we demonstrate that the NEL T3E RipAW interacts with FLS2, XLG2, and BIK1, three components of the immune receptor complex [7], and reduces their protein accumulation. Considering the E3 ligase activity of RipAW, we further showed that RipAW ubiquitinates FLS2, XLG2, and BIK1, consequently accelerating their degradation through the 26S proteasome pathway. These results preliminarily uncover the virulence mechanism of RipAW and more importantly, revealed for the first time the strategy of *R. solanacearum* to use type III effectors to target the immune receptor complex and eventually promote its infection in host plants. The partial rescue of the RipAW-inhibited ROS burst by the expression of FLS2, XLG2, or BIK1 indicates the involvement of the immune receptor complex in RipAW’s virulence function. However, the incomplete rescue of ROS burst may also suggest the potential involvement of other pathways in RipAW-manipulated host PTI. In-depth studies are worthy of being carried out in the future to further unlock the virulence mechanism of RipAW. We have also found that, besides the immune receptor complex, RipAW also interacted with some transcription factors and other membrane proteins, providing a new direction for our future research.

In summary, our results confirmed the virulence function of *R. solanacearum* T3E RipAW and demonstrated that this effector promotes *R. solanacearum* infection dependent on its E3 ligase activity. RipAW mediates the degradation of the immune receptor complex via the 26S proteasome to suppress host PTI, providing new insights into the pathogenicity mechanism of *R. solanacearum*.

## 4. Materials and Methods

### 4.1. Plant Materials and Growth Conditions

*RipAW*- and *RipAW^C177A^*- transgenic Arabidopsis (*Arabidopsis thaliana*) were generated using the floral dip method in the Col-0 background. The pGWB505-RipAW and pGWB505-RipAW^C177A^ vectors were used for the transgenic plants. All primers used for vector construction are listed in Appendix A. *GFP*-transgenic plants served as control plants. Seeds were surface sterilized with NaClO (30%, *v*/*v*) for 10 min, washed with sterilized water five times, stored in a 4 °C refrigerator in the dark for 2 days, and then sowed on 1/2 MS plates. The seeds were germinated at 22 °C under 16,000 lux with a photoperiod of 16 h light/8 h dark in a growth chamber for 10 days. Subsequently, the seedlings were transferred to soil and grown at 22 °C under 16,000 lux with a photoperiod of 10 h light/14 h dark in the greenhouse for an additional 3 weeks before assays. *Nicotiana benthamiana* seeds were sowed directly in the soil, kept moisture for five days, and then grown at 22 °C under 16,000 lux with a photoperiod of 16 h light/8 h dark for 4 weeks before experiments.

### 4.2. Pathogen Inoculation Assays

For soil-drenching inoculation assays with *Ralstonia solanacearum* GMI1000, one-week-old transgenic and control plants were transplanted into Jiffy Pots and grown at 22 °C under 16,000 lux with a photoperiod of 10 h light/14 h dark for 3 weeks. Each group comprised at least 12 plants. Soil-drenching inoculation assays were performed with a bacterial suspension containing 10^8^ colony-forming units per mL (CFU mL^−1^). After inoculation with 25 mL bacterial suspension per plant, plants were cultivated at 26 °C under 16,000 lux and 75% humidity with a photoperiod of 16 h light/8 h dark. Scoring of visual disease symptoms, based on a scale ranging from ‘0′ (no symptoms) to ‘4′ (complete wilting), was performed as previously described [38,39].

For the *Pseudomonas syringae* DC3000 and DC3000 *HrcC^−^* inoculation assays, strains were revived overnight, suspended with 10 mM MgCl_2_ containing 10^6^ colony-forming units per mL, and infiltrated into 4-week-old *A. thaliana* leaves with a sterile syringe. Bacterial populations were determined at 2 days post-inoculation. For the flg22 priming assay, seedlings were pre-treated with 1 μM flg22 for 12 h and then inoculated with *P. syringae* DC3000 as described above.

The seedling Petri-dish inoculation assay was performed as previously described [26]. One-week-old transgenic and control plants on 1/2 MS containing 0.8% agar were transplanted to the prepared 1/2 MS Petri dish containing 1.5% agar with *R. solanacearum* GMI1000 or GMI1000 *HrcV*^−^. Bacterial populations were determined at 3 dpi.

### 4.3. Agrobacterium-Mediated Transient Expression and Flg22-Induced ROS Burst

*Agrobacterium*-mediated transient expression was performed in five-week-old *N. benthamiana*, according to Ouyang et al. [25]. Flg22-induced ROS burst was monitored as previously described [8].

### 4.4. Co-Immunoprecipitation (co-IP)

Leaf tissue (0.5 g) was collected after infiltration with *A. tumefaciens* and immediately frozen in liquid nitrogen to perform co-immunoprecipitation. Total proteins were extracted with extraction buffer (10% *v/v* Glycerol, 25 mM Tris-HCl, 1 mM EDTA, 150 mM NaCl, 10 mM DTT, 1 mM PMSF, 0.5% Triton X-100, 1 × Protease Inhibitor Cocktail). Then, total proteins co-immunoprecipitated with 10 μL ANTI-FLAG M2 Affinity Agarose Gel (Sigma, St. Louis, MO, USA) for 2 h after equilibrating with extraction buffer 3 times. The beads were then washed 5 times with wash buffer (25 mM Tris-HCl, 1 mM EDTA, 150 mM NaCl, 1 mM PMSF, 0.5% Triton X-100). Immunoprecipitated proteins were stripped from beads by boiling in 40 μL Laemmli buffer for 10 min at 100 °C. The immunoprecipitated proteins were separated with 12% SDS-PAGE gels for western blot with different antibodies to detect the interaction between different proteins.

### 4.5. Split-Luciferase Complementation (LCA) Assay

LCA was performed as previously described [40] with slight modifications. Briefly, *A. tumefaciens* GV3101 containing the different plasmids was infiltrated into *N. benthamiana* leaves for 36 h. Subsequently, a 1 mM luciferin solution was infiltrated into the same area. Each whole leaf was detached from the plant and kept in the dark for 10 min before imaging to avoid interference by chlorophyll luminescence. Luminescence images were captured using SH-Compact 523 (Shenhua Science Technology, Hangzhou, China) with SHST capture software (version 2.0.1.107). The combination of BIK1-nLUC and XLG2-cLUC was used as a positive control, while that of BIK1-nLUC and CPR5-cLUC served as a negative control.

### 4.6. Bimolecular Fluorescence Complementation (BiFC) Assay

The coding sequences of FLS2, BIK1, and XLG2 were separately fused with nYFP, and the coding sequence of RipAW was fused with cYFP, as previously described [41]. The recombinant vector was verified by sequencing. The plasmids were transferred into *A. tumefaciens* GV3101 and infiltration of *N. benthamiana* was performed as described above. Infiltrated tissues were visualized to assess the interaction of FLS2-nYFP, BIK1-nYFP, or XLG2-nYFP with RipAW-cYFP at approximately 36 h after infiltration. Fluorescence was observed using a Leica TCS SP8 X White Light Laser Confocal Microscope (Leica, Wetzlar, Germany), and images were superimposed using the LAS X Life Science Microscope Software Platform (version 3.1.5.16308).

### 4.7. Quantitative Real-Time PCR

Total RNA was isolated using Trizol reagent (Sangon, Shanghai, China), according to the manufacturer’s recommendations. The expression of PTI-related genes *WRKY29* and *FRK1* [42] was analyzed by quantitative real-time PCR as previously described [25]. The *ACTIN* gene was used as an internal control. Gene-specific RT-PCR primers were designed based on their cDNA sequences and listed in Appendix A.

### 4.8. Callose Deposition Assay

The callose deposition assay was performed as previously reported [43] with minor modifications. Briefly, rosette leaves of four-week-old plants were pre-infiltrated with 1 mM of flg22 for 12 h, sampled to decolorate with ethanol overnight, and rehydrated with 75% ethanol, 50% ethanol, and sterile water successively. Leaves were then incubated with stain buffer (0.01% aniline blue, 150 mM K_2_HPO_4_ pH 9.5) for 2 h at room temperature in the dark. Callose deposition was detected using an Axio Imager M2 microscope (Zeiss, Oberkochen, Germany) fluorescence microscope, and the number of callose deposits was counted with ImageJ software (version 1.49).

### 4.9. In vivo Ubiquitination Assay

The in vivo ubiquitination assay was performed as previously described [44] with minor modifications. We co-expressed FLS2, XLG2, or BIK1 with RipAW, RipAW^C177A^, or LTI6b, respectively. The infiltrated tissue was then treated with MG132 at 36 hpi and sampled after an additional 8 h. Samples were immediately frozen in liquid nitrogen before the in vivo ubiquitination assay. Total proteins were extracted and subjected to an immunoprecipitation assay using anti-Flag beads. The ubiquitination levels were detected by an anti-ubiquitin antibody (Jingjie PTM BioLab, Hangzhou, China), while the loading proteins were detected by anti-FLAG (Abclonal Technology, Wuhan, China), anti-HA (Abclonal Technology, Wuhan, China), or anti-MYC antibody (Abmart, Shanghai, China).

### 4.10. Statistical Analysis

All experiments were repeated at least three times with similar results. Data were analyzed and statistically compared using Student’s *t*-test along with analysis of variance.

## Figures and Tables

**Figure 1 ijms-25-00183-f001:**
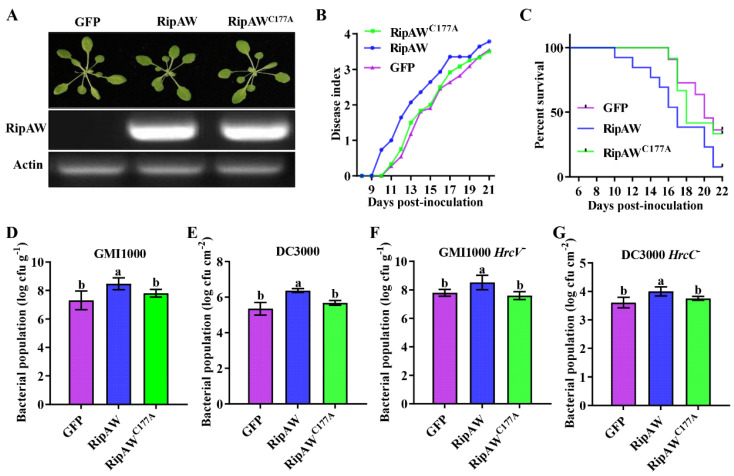
Generation of *RipAW*- and *RipAW*^C177A^-transgenic *Arabidopsis thaliana* and evaluation of disease resistance in these transgenic plants. (**A**) The growth phenotype of *RipAW*- and *RipAW^C177A^*-transgenic plants and the expression of transgenes verified by RT-PCR. (**B**,**C**) The disease index (**B**) and survival rate (**C**) of *GFP*-, *RipAW*-, and *RipAW^C177A^*-transgenic plants that were soil-drenching inoculated with GMI1000. (**D**) The bacterial population in plants that were Petri-dish inoculated with GMI1000. (**E**) The bacterial population in plants that were leaf-inoculated with DC3000. (**F**) The bacterial population in plants that were Petri-dish inoculated with GMI1000 *HrcV^−^*. (**G**) The bacterial population in plants that were leaf-inoculated with DC3000 *HrcC^−^*. Values are means ± standard errors (SEs) (n = 12 for (**B**,**C**), and n = 6 for (**D**–**G**)). Different lowercase letters indicate significant differences (*p* < 0.05).

**Figure 2 ijms-25-00183-f002:**
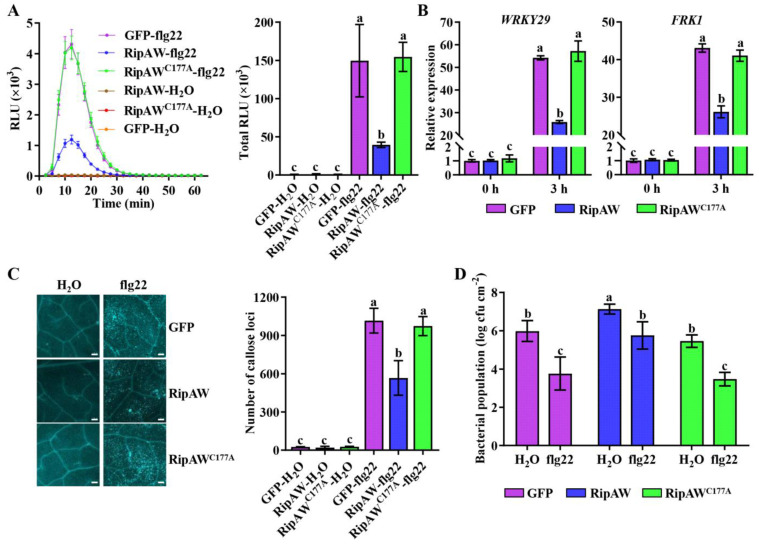
RipAW impairs PTI responses in *A. thaliana*. (**A**) The time course curve of flg22-induced ROS burst and total ROS accumulation in *RipAW*-, *RipAW^C177A^*- and *GFP*-transgenic plants. (**B**) Relative expression levels of PTI marker genes, *WRKY29* and *FRK1*, in 4-week-old *RipAW*-, *RipAW^C177A^*-, and *GFP*-transgenic plants. Leaves were treated with 1 μM flg22 for 3 h, and gene expression levels at 0 h and 3 h were measured by qRT-PCR. *ACTIN* was used as an internal reference gene. (**C**) Callose deposition levels of 4-week-old *RipAW*-, *RipAW^C177A^*-, and *GFP*-transgenic plants. Leaves were infiltrated with 1 μM flg22 at 12 h before sampling. Scale bars represent 200 μm. (**D**) The defense priming assay in *RipAW*-, *RipAW^C177A^*-, and *GFP*-transgenic plants. Four-week-old plants were pre-treated with 1 μM flg22 12 h before *P. syringae* DC3000 inoculation. Bacterial population was determined at 2 days post-inoculation. Values are means ± SEs (n = 6). Different lowercase letters indicate significant differences (*p* < 0.05).

**Figure 3 ijms-25-00183-f003:**
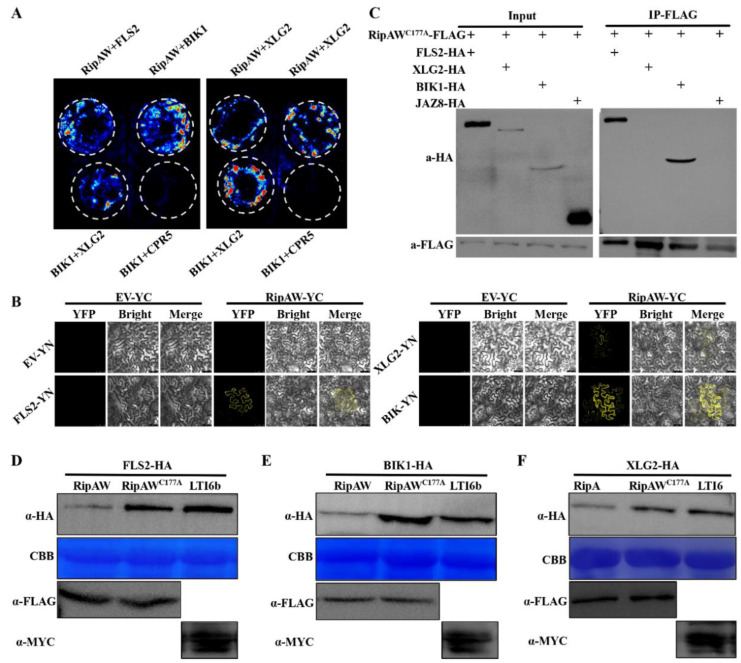
RipAW associates with the immune receptor complex and decreases the protein accumulation of FLS2, XLG2, and BIK1. (**A**) Split-luciferase complementation assay evaluating the association of RipAW-nLUC with XLG2-cLUC, FLS2-cLUC, or BIK1-cLUC. The combination of BIK1-nLUC and XLG2-cLUC served as a positive control, while the combination of BIK1-nLUC and CPR5-cLUC acted as a negative control. (**B**) Bimolecular fluorescence complementation assay detecting the interaction between RipAW-YC and FLS2-YN, XLG2-YN, or BIK1-YN. The combination of EV-YN and EV-YC was used as a negative control. YFP indicated the YFP fluorescence and bright-field images showed the plant cell structure. The merged images showed the overlay of YFP fluorescence on the bright field. (**C**) Association of RipAW with the immune receptor complex detected by co-immunoprecipitation assay. FLS2-HA, XLG2-HA, or BIK1-HA was co-expressed with RipAW-FLAG in 5-week-old *N. benthamiana* leaves. Samples were collected at 36 h after infiltration, and the protein extracts were subjected to co-immunoprecipitation using anti-FLAG beads. The protein expression levels were checked in the input. The association between RipAW and FLS2, XLG2, or BIK1 was detected by an anti-HA antibody in immunoprecipitation. (**D**) FLS2 protein accumulation; (**E**) XLG2 protein accumulation; (**F**) BIK1 protein accumulation. FLS2-HA, XLG2-HA, or BIK1-HA was co-expressed with RipAW-FLAG, RipAW^C177A^-FLAG, or LTI6b-MYC in 5-week-old *N. benthamiana* leaves. Samples were collected at 36 h after infiltration.

**Figure 4 ijms-25-00183-f004:**
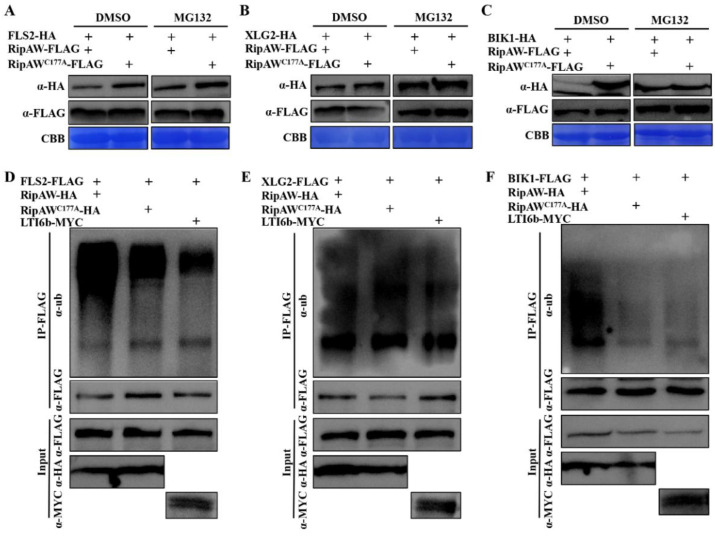
RipAW accelerates degradation of FLS2, XLG2, and BIK1 through the 26S proteasome pathway. (**A**–**C**) Protein accumulation of FLS2 (**A**), XLG2 (**B**), and BIK1 (**C**) co-expressed with RipAW or RipAW^C177A^ under DMSO or MG132 treatment. (**D**–**F**) Ubiquitination levels of FLS2 (**D**), XLG2 (**E**), and BIK1 (**F**) when co-expressed with RipAW, RipAW^C177A^, or LTI6b under MG132 treatment.

**Figure 5 ijms-25-00183-f005:**
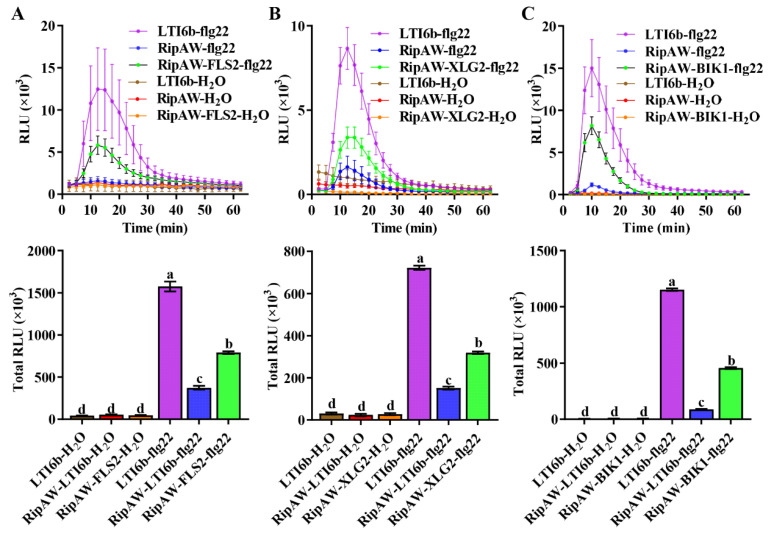
Expression of FLS2, XLG2, or BIK1 partially reverses the defect in ROS burst caused by RipAW in *N. benthamiana.* (**A**) Effect of FLS2 expression on ROS burst. (**B**) Effect of BIK1 expression on ROS burst. (**C**) Effect of XLG2 expression on ROS burst. Different lowercase letters indicate significant differences (*p* < 0.05).

## Data Availability

Data is contained within the article.

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
