# Peer review of "The Ralstonia solanacearum Type III Effector RipAW Targets the Immune Receptor Complex to Suppress PAMP-Triggered Immunity"

_ijms, 2023, doi:10.3390/ijms25010183_

Round 1

Reviewer 1 Report

Comments and Suggestions for Authors

The submitted manuscript to IJMS entitled " Ralstonia solanacearum Type III Effector RipAW Targets Immune Receptor Complex to Suppress PAMP-Triggered Immunity” has the potential to be published in IJMS. But, before acceptance, there are few comments that need to be addressed by the authors:

 The English language of the MS needs to be improved.

In the introduction section, the research gap is not mentioned, and the objectives are not clearly mentioned as well.

The discussion section should be extended.

The conclusion section should be re-written.

Comments on the Quality of English Language

Author Response

I appreciate the valuable suggestions provided by the reviewer to improve our manuscript. We have diligently revised our manuscript, and we believe it now aligns with publication standards of IJMS. The details of the revision are outlined below:

  1. The manuscript has undergone language revisions by an expert in our field, and the modifications are highlighted in blue throughout the main text.
  2. Objectives of the research have been included in the Introduction section, addressing the reviewer's comments.
  3. A more extensive discussion has been added to elucidate the partial rescue of ROS by the components of the immune receptor complex.
  4. The conclusion section has been rephrased for greater conciseness.

We believe that these revisions have significantly enhanced the quality of our manuscript. Thanks again for your efforts to improve our paper.

Reviewer 2 Report

Comments and Suggestions for Authors

This meticulously crafted manuscript delves into the pathogenesis of Ralstonia solanacearum, a pivotal pathogen that concurrently serves as a crucial research model for understanding its host's response. The experimental design exhibits meticulous planning and thoughtful consideration. The obtained results are unequivocal, leaving no room for doubt. The authors convincingly demonstrate that RipAW engages in the ubiquitination of FLS2, XLG2, and BIK1, unequivocally substantiating its role in accelerating their degradation through the 26S proteasome pathway. This significantly contributes to the body of knowledge surrounding the pathogenesis of Ralstonia solanacearum

Author Response

I sincerely appreciate the reviewer's positive comments on our manuscript. Thank you very much.